# GEMINI: Controlling The Sentence-Level Summary Style in Abstractive Text Summarization

**Guangsheng Bao[1,2], Zebin Ou[2], and Yue Zhang*[2,3],**

[1] Zhejiang University
[2] School of Engineering, Westlake University
[3] Institute of Advanced Technology, Westlake Institute for Advanced Study
[2] {baoguangsheng, ouzebin, zhangyue}@westlake.edu.cn

## Abstract

Human experts write summaries using different techniques, including extracting a sentence from the document and rewriting it, or fusing various information from the document to abstract it. These techniques are flexible and thus difficult to be imitated by any single method. To address this issue, we propose an adaptive model, GEMINI, that integrates a rewriter and a generator to mimic the sentence rewriting and abstracting techniques, respectively. GEMINI adaptively chooses to rewrite a specific document sentence or generate a summary sentence from scratch. Experiments demonstrate that our adaptive approach outperforms the pure abstractive and rewriting baselines on three benchmark datasets, achieving the best results on WikiHow. Interestingly, empirical results show that the human summary styles of summary sentences are consistently predictable given their context. We release our code and model at https://github.com/baoguangsheng/gemini.

## 1 Introduction

Text summarization aims to automatically generate a fluent and succinct summary for a given text document (Maybury, 1999; Nenkova and McKeown, 2012; Allahyari et al., 2017). Two dominant methods have been used, namely extractive and abstractive summarization techniques. *Extractive methods* (Nallapati et al., 2017; Narayan et al., 2018; Liu and Lapata, 2019; Zhou et al., 2020; Zhong et al., 2020) identify salient text pieces from the input and assemble them into an output summary, leading to faithful but possibly redundant and incoherent summaries (Chen and Bansal, 2018; Gehrmann et al., 2018; Cheng and Lapata, 2016), where rewriting techniques could be used to further reduce redundancy and increase coherence (Bae et al., 2019; Bao and Zhang, 2021). In contrast, *abstractive methods* (Rush et al., 2015; Nallapati et al., 2016;

---
\* Corresponding author.

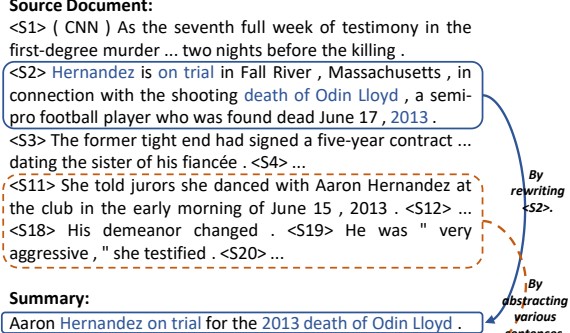

**Source Document:**
<S1> ( CNN ) As the seventh full week of testimony in the first-degree murder ... two nights before the killing .
<S2> Hernandez is on trial in Fall River , Massachusetts , in connection with the shooting death of Odin Lloyd , a semi-pro football player who was found dead June 17 , 2013 .
<S3> The former tight end had signed a five-year contract ... dating the sister of his fiancée . <S4> ...
<S11> She told jurors she danced with Aaron Hernandez at the club in the early morning of June 15 , 2013 . <S12> ...
<S18> His demeanor changed . <S19> He was " very aggressive , " she testified . <S20> ...

By rewriting <S2>.

By abstracting various sentences.

**Summary:**
Aaron Hernandez on trial for the 2013 death of Odin Lloyd . Jurors hear of defendant 's demeanor at club .

Figure 1: Example of human summary from CNN/DM.

See et al., 2017; Lewis et al., 2020) apply natural language generation (NLG) technologies in synthesizing the output, which obtains more concise and coherent summaries, but at the cost of degraded faithfulness (Huang et al., 2020; Maynez et al., 2020).

The effectiveness of extractive and abstractive methods relies on the summary style of summaries. Study shows that human take different styles in writing each summary sentence (Jing and McKeown, 1999), which we categorize broadly into extractive and abstractive styles. *Extractive style* mainly conveys ideas directly from article sentences, while *abstractive style* conveys new ideas that are entailed from various article sentences. One example is shown in Figure 1, where the summary consists of two sentences. The first sentence is generated by rewriting sentence <S2> from the document. In contrast, the second summary sentence is generated by abstracting various sentences. These styles are flexible and thus difficult to be imitated by a single method.

In this paper, we aim to mimic human summary styles, which we believe can increase our ability to control the style and deepen our understanding of how human summaries are produced. To better adapt to summary styles in human summaries, we propose an adaptive model, **GEMINI**, which

contains a rewriter and a generator to imitate the extractive style and the abstractive style, respectively. For the *rewriter*, we adopt contextualized rewriting (Bao and Zhang, 2021) integrated with an inner sentence extractor. For the *generator*, we use standard seq2seq summarizer (Lewis et al., 2020).

The rewriter and the generator are integrated into a single decoder, using a *style controller* to switch the style of a generation. The style controller assigns different group tags so that relevant sentences in the input can be used to guide the decoder. In the abstractive style, the decoder does not focus on a specific sentence, while in the extractive style, the decoder is guided by attending more to a particular rewritten sentence. In order to train such an adaptive summarizer, we generate oracle extractive/abstractive styles using automatic detection of sentence-level summary styles.

We evaluate our model on three representative benchmark datasets. Results show that GEMINI allows the model to better fit training data thanks to differentiating summary styles. Our adaptive rewriter-generator network outperforms the strong abstractive baseline and recent rewriter models with a significant margin on the benchmarks. Interestingly, experiments also show that the summary style of a sentence can be consistently predicted during test time, which shows that there is underline consistency in the human choice of summary style of a summary sentence in context. To our knowledge, we are the first to explicitly control the style per summary sentence, and at the same time, achieve improved ROUGE scores. Our automatic style detection metric allows further quantitative analysis of summary styles in the future.

## 2  Related Work

Abstractive summarizers achieve competitive results by fine-tuning on pre-trained seq2seq model (Lewis et al., 2020; Zhang et al., 2020). Our work is in line, with the contribution of novel style control over the summary sentence, so that a model can better fit human summaries in the "sentence rewriting" and "long-range abstractive" styles. Our generator is a standard seq2seq model with the same architecture as BART (Lewis et al., 2020), and our rewriter is related to previous single sentence rewriting (Chen and Bansal, 2018; Bae et al., 2019; Xiao et al., 2020) and contextualized rewriting (Bao and Zhang, 2021).

Our rewriter uses the contextualized rewriting

| Summary Style | CNN/DM | XSum | WikiHow |
|---|---|---|---|
| Extractive | 88.6% | 11.8% | 61.1% |
| Abstractive | 11.4% | 88.2% | 38.9% |

Table 1: *Human evaluation* of sentence-level summary style on three datasets. The percentage denotes the ratio of summary sentences belonging to the style.

mechanism, which considers the document context of each rewritten sentence, so that important information from the context can be recalled and cross-sentential coherence can be maintained. However, different from Bao and Zhang (2021), which relies on an external extractor to select sentences, we integrate an internal extractor using the pointer mechanism (Vinyals et al., 2015), analogous to NeuSum (Zhou et al., 2020) to select sentences autoregressively. To our knowledge, we are the first to integrate a rewriter and a generator into a standalone abstractive summarizer.

Our model GEMINI can also be seen as a mixture of experts (Jacobs et al., 1991), which dynamically switches between a rewriter and a generator. A related work is the pointer-generator network (See et al., 2017) for seq2seq modeling, which can also viewed as a mixture-of-expert model. Another related work is the HYDRASUM, which has two experts for decoding. These models can be viewed as a *soft* mixture of experts, which learns latent experts and makes decisions by integrating their outputs. In contrast, our model can be viewed as a *hard* mixture of experts, which consults either the rewriter or the generator in making a decision. A salient difference between the two models is that our GEMINI makes decisions for each *sentence*, while previous work makes decisions at the *token* level. The goals of the models are very different.

## 3  Summary Style at Sentence Level

### 3.1  Human Evaluation of Summary Style

We conduct a human evaluation of the summary style of each summary sentence, annotating as *extractive style* if the summary sentence can be implied by one of the article sentences, and *abstractive style* if the summary sentence requires multiple article sentences to imply. We sample 100 summary sentences for each dataset and ask three annotators to annotate them, where we take the styles receiving at least two votes as the final labels.

The averaged distributions of styles are shown in Table 1. CNN/DM is mostly extractive style, which has 88.6% summary sentences written in the extractive style. In contrast, XSum is mostly

| Dataset | Novel n-grams | | | Fragment | | Ours |
|---|---|---|---|---|---|---|
| | 1-gram | 2-gram | 3-gram | Coverage | Density | Fusion Index |
| CNN/DM | 0.62 | 0.55 | 0.56 | 0.59 | 0.56 | **0.76** |
| XSum | 0.25 | 0.17 | 0.24 | 0.15 | 0.16 | **0.46** |
| WikiHow | 0.47 | 0.49 | 0.46 | 0.31 | 0.30 | **0.56** |

Table 2: Our automatic metrics compare with previous metrics in measuring fusion degrees. The number shows the Pearson correlation between the metric and human-annotated fusion degrees.

abstractive style, which has 88.2% summary sentences written in the abstractive style. WikiHow has a more balanced distribution of summary styles, with about 60% summary sentences in the extractive style. The results suggest that real summaries are a mixture of styles, even for the well-known extractive dataset CNN/DM and abstractive dataset XSum.

## 3.2 Automatic Detection of Summary Style

Because of the high cost of human annotation, we turn to automatic approaches to detect the summary styles at the sentence level. Previous studies mainly measure the summary style at the summary level. For example, Grusky et al. (2018) propose the *coverage* and *density* of extractive fragments to measure the extractiveness. See et al. (2017) propose the proportion of *novel n-grams*, which appear in the summary but not in the input document, as indicators of abstractiveness. We adopt these metrics to sentences and take them as baselines of our approach.

We propose *fusion index* to measure the degree of fusing context information (Barzilay and McKeown, 2005), considering two factors: 1) How much information from the summary sentence can be recalled from one document sentence? If the recall is high, the sentence is more likely to be extractive style because it can be produced by simply rewriting the document sentence. 2) How many document sentences does the summary sentence relate to? A larger number of such sentences indicates a higher fusion degree, where the summary sentence is more likely to be abstractive style. Our fusion index is calculated on these two factors.

**Recall.** We measure the percentage of recallable information by matching the summary sentence back to document sentences. Given a summary sentence $S$ and a source document $D = \{S_1, S_2, ..., S_{|D|}\}$, we find the best-match

$$RC(S \mid D) = \max_{1 \le i \le |D|} R(S \mid S_i), \quad (1)$$

where $R(S|S_i)$ is an average of ROUGE-1/2/L recalls of $S$ given $S_i$, representing the percentage of information of sentence $S$ covered by sentence $S_i$.

**Scatter.** We measure the scattering of the content of a summary sentence by matching the sentence to all document sentences. If the matching scores are equally distributed on all document sentences, the scatter is high; If the matching scores are all zero except for one document sentence, the scatter is low. We calculate the scatter using the distribution entropy derived from the matching scores.

$$SC(S \mid D) = -\sum_{j=1}^{K} p_j \log p_j / \log K, \quad (2)$$

where $p_j$ is the estimated probability whether $S$ is generated from the corresponding document sentence $S_i$, calculated using the top-$K$ best-matches $\{r_j\}|_{j=1}^{K}$ that

$$p_j = r_j / \sum_{j=1}^{K} r_j,$$
$$\{r_j\}|_{j=1}^{K} = \text{Top}(\{R(S|S_i) \mid 1 \le i \le |D|\}, K). \quad (3)$$

The hyper-parameter $K$ is determined using an empirical search on the human evaluation set.

**Fusion index.** We calculate fusion index (FI) from *Recall (RC)* and *Scatter (SC)*

$$FI(S \mid D) = (1 - RC(S \mid D)) * SC(S \mid D), \quad (4)$$

which represents the degree of fusion, where 0 means no fusion, 1 means extreme fusion, and others between.

We *evaluate* our metrics together with the candidates from previous studies, reporting the Pearson correlation with human-annotated summary styles as shown in Table 2. Our proposed fusion index gives the best correlation with summary styles. In contrast, among previous metrics, only the novel 1-gram has the closest correlation but is still lower than the fusion index by about 0.14 on average. The results suggest that the fusion index is a more suitable extractive-abstractive measure at the sentence level.

## 3.3 Oracle Label for Summary Style

We produce oracle extractive/abstractive labels using automatic fusion index so that we can train a

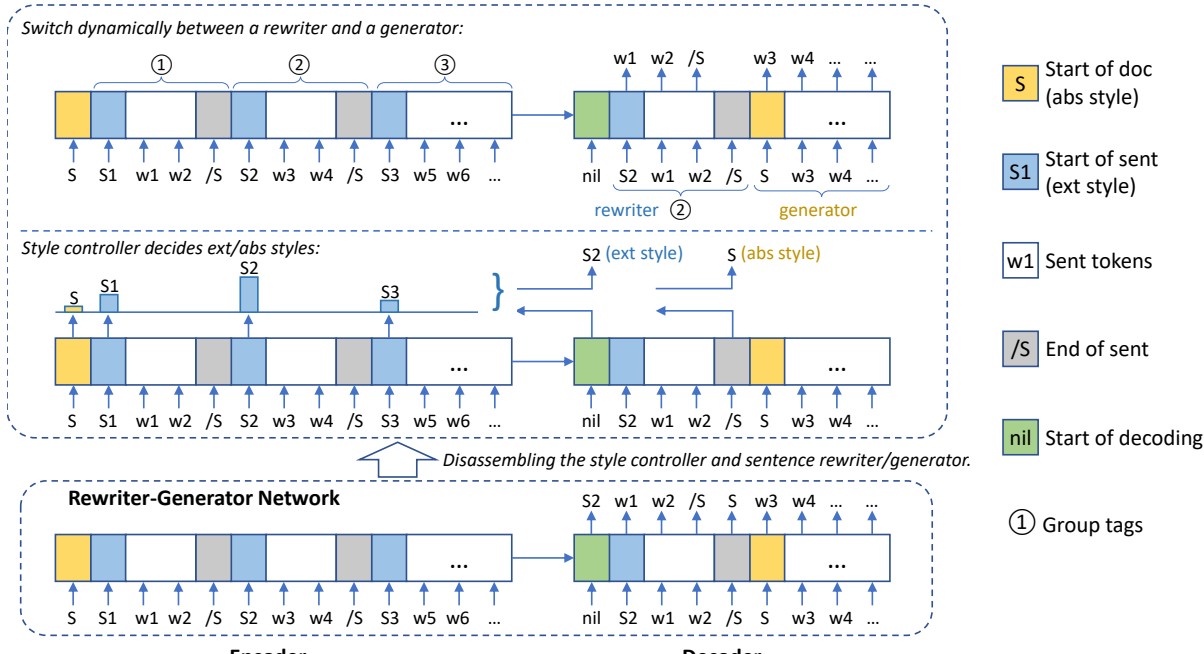

Figure 2: GEMINI uses a controller to decide ext/abs styles, and further switch the decoder accordingly between a rewriter and a generator. We express the identifier tokens "" and "<$S_k$>" in the concise form "S" and "$S_k$", and we use "w1 w2" to represent the tokens in the first sentence and "w3 w4" the tokens in the second sentence. Group tags are converted into embeddings and added to the input token embeddings for both the encoder and decoder. The decoder predicts an identifier token to determine the group tag of the following timesteps, for example "<S2>" to start the group tag "2" until the end of the sentence "".

summarizer with explicit style control at the sentence level. If the fusion index is higher than a threshold, we treat the sentence as extractive style. If the fusion index of the summary sentence is lower than the threshold, we treat the sentence as abstractive style. We search for the best threshold for each dataset using development experiments.

## 4 GEMINI: Rewriter-Generator Network

As Figure 2 shows, GEMINI adopts pre-trained BART (Lewis et al., 2020), using a *style controller* to decide the summary style. According to the style, either the *rewriter* or the *generator* is activated to generate each sentence.

Intuitively, GEMINI performs the best when it works on a dataset with a balanced style, which enables the rewriter and generator to complement each other, and when the oracle styles are of high accuracy so that the training supervision can be of high quality.

### 4.1 Input and Output

We introduce special identifier tokens, including "" - the start of a document, "<$S_k$>" - the start of a sentence $k$, and "" - the end of a sentence.

We express the input document as " <$S_1$> *sentence one* .  <$S_2$> *sentence two* .  <$S_3$> *sentence three* .  ...*", where the sequence starts with the identifier token "" and enclose each sentence with "<$S_k$>" and "". We express the output summary as "<$S_2$> *sentence one* .   *sentence two* .  ...*", where each sentence starts with "<$S_k$>" or "" and ends with "". If a summary sentence starts with "<$S_k$>", the decoder will generate the sentence according to the $k$-th document sentence. If a summary sentence starts with "", the decoder will generate the sentence according to the whole document.

### 4.2 Document Encoder

We extend the embedding table to include the identifier tokens so that the *style controller* can match their embeddings to decide the summary style. We follow contextualized rewriting (Bao and Zhang, 2021) to allocate a group tag to each input sentence so that the *rewriter* can rely on these group tags to locate the rewritten sentence using multi-head attention, as the group tags ① ② ③ in Figure 2 illustrate.

Specifically, the first summary sentence and the second document sentence have the same group

| Dataset | Source | Train | Valid | Test | Document | | Summary | | |
|---------|--------|-------|-------|------|----------|---|---------|---|---|
| | | | | | Words | Sents | Words | Sents | #W/S |
| CNN/DM | News | 287,227 | 13,368 | 11,490 | 788 | 36.9 | 56 | 3.8 | 15 |
| XSum | News | 204,045 | 11,332 | 11,334 | 431 | 18.7 | 23 | 1.0 | 23 |
| WikiHow | Knowledge Base | 168,128 | 6,000 | 6,000 | 584 | 32.2 | 62 | 7.5 | 8 |

Table 3: Datasets for evaluation. The number of words, the number of sentences, and the number of words per sentence (#W/S) are all average values.

tag of ②, which are converted to group-tag embeddings and added to the token embeddings in the sentence. Using a shared group-tag embedding table between the encoder and decoder, the decoder can be trained to concentrate on the second document sentence when generating the first summary sentence.

Formally, we express the group-tag embedding table as $\text{EMB}_{tag}$ and generate a group-tag sequence $G_X$ uniquely from $X$ according to the index of each sentence as

$$G_X = \{k \text{ if } w_i \in \text{S}_k \text{ else } 0\}|_{i=1}^{|X|}, \qquad (5)$$

where for a token $w_i$ in the $k$-th sentence $\text{S}_k$ we assign a group tag of number $k$. We convert $G_X$ into embeddings and inject it in the BART encoder.

We take the standard Transformer encoding layers, which contain a self-attention module and a feed-forward module:

$$x^{(l)} = \text{LN}(x^{(l-1)} + \text{SELFATTN}(x^{(l-1)})),$$
$$x^{(l)} = \text{LN}(x^{(l)} + \text{FEEDFORWARD}(x^{(l)})), \qquad (6)$$

where LN denotes layer normalization (Ba et al., 2016). The last layer $L$ outputs the final encoder output $x_{out} = x^{(L)}$. The vectors $x_{emb}$ and $x_{out}$ are passed to the decoder for prediction.

### 4.3 Summary Decoder

We extend the BART decoder with a style controller and a rewriter, while for the generator, we use the default decoder.

**Style Controller.** We use attention pointer (Vinyals et al., 2015) to predict the styles, where we weigh each input sentence (attention on "<$\text{S}_k$>") or the whole document (attention on ""). If "" receives the largest attention score, we choose the abs style, and if <$\text{S}_k$> receives the most attention score, we choose the ext style.

At the beginning of the summary or at the end of a summary sentence, we predict the style of the next summary sentence. We match the token output to the encoder outputs to decide the selection.

$$y_{match} = y_{out} \times (x_{out} * \alpha + x_{emb} * (1 - \alpha))^T, \qquad (7)$$

where $\alpha$ is a trainable scalar to mix the encoder outputs $x_{out}$ and token embeddings $x_{emb}$. We match these mix embeddings to the decoder outputs $y_{out}$, obtaining the logits $y_{match}$ of the pointer distribution. We only keep the logits for sentence identifiers, including "" and "<$\text{S}_k$>", and predict the distribution using a softmax function.

**Rewriter and Generator.** We use the standard decoder as the backbone for both the rewriter and the generator. For the rewriter, we follow Bao and Zhang (2021) to apply group-tag embeddings to the input of the decoder. For the generator, we do not apply group-tag embeddings, so that it does not correspond to any document sentence.

Formally, given summary $Y = \{w_j\}|_{j=1}^{|Y|}$, we generate $G_Y$ uniquely from $Y$ according to the identifier token "<$\text{S}_k$>" and "" that for each token $w_i$ in the sentence started with "<$\text{S}_k$>" the group-tag is $k$ and for that with "" the group-tag is 0. For instance, if $Y$ is the sequence "<$\text{S}_2$> $w_1 w_2$   $w_3 w_4$  <$\text{S}_7$> ...", $G_Y$ will be $\{2, 2, 2, 2, 0, 0, 0, 0, 7, ...\}$.

We do not change the decoding layers, which contain a self-attention module, a cross-attention module, and a feed-forward module for each:

$$y^{(l)} = \text{LN}(y^{(l-1)} + \text{SELFATTN}(y^{(l-1)})),$$
$$y^{(l)} = \text{LN}(y^{(l)} + \text{CROSSATTN}(y^{(l)}, x_{out})), \qquad (8)$$
$$y^{(l)} = \text{LN}(y^{(l)} + \text{FEEDFORWARD}(y^{(l)})),$$

where LN denotes layer normalization (Ba et al., 2016). The last layer $L$ outputs the final decoder output $y_{out}$. The decoder output $y_{out}$ is then matched with token embeddings for predicting the next tokens.

### 4.4 Training and Inference

We use MLE *loss* for both token prediction and style prediction. We calculate the overall loss as

$$\mathcal{L} = \mathcal{L}_{token} + \kappa * \mathcal{L}_{style}, \qquad (9)$$

where $\mathcal{L}_{style}$ is the MLE loss of the style prediction, and $\mathcal{L}_{token}$ is the MLE loss of the token prediction. Because of the different nature between style and

| Dataset | Method | R-1 | R-2 | R-L |
|---------|--------|-----|-----|-----|
| CNN/DM | BRIO (Liu et al., 2022) | 47.78 | 23.55 | 44.57 |
| | SimCLS (Liu and Liu, 2021) | 46.67 | 22.15 | 43.54 |
| | GSum (Dou et al., 2021) | 45.94 | 22.32 | 42.48 |
| | PEGASUS (Zhang et al., 2020) | 44.17 | 21.47 | 41.11 |
| | GOLD (Pang and He, 2020) | 45.40 | 22.01 | 42.25 |
| | BERT-Rewriter (Bao and Zhang, 2021) | 43.52 | 20.57 | 40.56 |
| | BERT+Copy/Rewrite (Xiao et al., 2020) | 42.92 | 19.43 | 39.35 |
| | BERT-Ext+Abs+RL (Bae et al., 2019) | 41.90 | 19.08 | 39.64 |
| | **Fair Settings** | | | |
| | BART-Rewriter (Bao and Zhang, 2023) | 44.26 | 21.23 | 41.34 |
| | BART (Lewis et al., 2020) | 44.16 | 21.28 | 40.90 |
| | GEMINI (Ours) | **45.27*** | **21.77*** | **42.34*** |
| XSum | BRIO (Liu et al., 2022) | 49.07 | 25.59 | 40.40 |
| | GSum (Dou et al., 2021) | 45.40 | 21.89 | 36.67 |
| | PEGASUS (Zhang et al., 2020) | 47.21 | 24.56 | 39.25 |
| | GOLD (Pang and He, 2020) | 45.85 | 22.58 | 37.65 |
| | **Fair Settings** | | | |
| | HYDRASUM (Goyal et al., 2022) | 44.61 | 20.91 | 36.17 |
| | BART (Lewis et al., 2020) | 45.14 | 22.27 | 37.25 |
| | GEMINI (Ours) | **45.86*** | **22.55*** | **37.68*** |
| WikiHow | RefSum (Liu et al., 2021) | 42.12 | 18.13 | 40.66 |
| | GSum (Dou et al., 2021) | 41.74 | 17.73 | 40.09 |
| | PEGASUS (Zhang et al., 2020) | 41.35 | 18.51 | 33.42 |
| | **Fair Settings** | | | |
| | BART (Lewis et al., 2020) | 41.46 | 17.80 | 39.89 |
| | GEMINI (Ours) | **42.43*** | **19.36*** | **41.12*** |

Table 4: *Automatic evaluation* on ROUGE scores, where * denotes the significance with a t-test of $p < 0.01$ compared to BART baseline. For a fair comparison, we list mainly the BART-based models as our baselines, leaving other pre-trained models, ensemble methods, and re-ranking techniques out of our scope for comparison.

token predictions, we use a hyper-parameter $\kappa$ to coordinate their convergence speed of them. In practice, we choose $\kappa$ according to the best performance on the development set.

During *inference*, the style controller is activated first to decide on an ext/abs style. If it chooses the ext style, the matched sentence identifier "$<S_k>$" will be used to generate group tags for the following tokens. If it chooses the abs style, we make the tokens with a special group tag of 0.

## 5 Experimental Settings

We use three English benchmark *datasets* representing different styles and domains as shown in Table 3.

**CNN/DailyMail** (Hermann et al., 2015) is the most popular single-document summarization dataset, comprising online news articles and human written highlights.

**XSum** (Narayan et al., 2018) is an abstractive style summarization dataset built upon new articles with a one-sentence summary written by professional authors.

**WikiHow** (Koupaee and Wang, 2018) is a diverse summarization dataset extracted from the online knowledge base WikiHow, which is written by human authors.

To generate oracle styles, we use fusion index thresholds of $\gamma = 0.7$, $\gamma = 0.7$, and $\gamma = 0.3$ for CNN/DM, XSum, and WikiHow, respectively.

We *train* GEMINI using a two-stage strategy: pre-finetuning the new parameters and then joint fine-tuning all the parameters. Since we introduce additional structure and new parameters to the pre-trained BART, direct joint fine-tuning of the two types of parameters can cause a downgrade of the pre-trained parameters. We introduce *pre-finetuning* to prepare the random-initialized parameters by freezing the pre-trained parameters and fine-tuning for 8 epochs before the joint fine-tuning. We use the same MLE loss for the two stages, using a convergence coordination parameter of $\kappa = 1.1$.

## 6 Results

### 6.1 Automatic Evaluation

As Table 4 shows, we evaluate our model on three benchmark datasets, in comparison with other BART-based models in fair settings. We report automatic metric ROUGE-1/2/L (Lin, 2004).

Compared to the abstractive BART baseline, GEMINI improves the ROUGE scores by an average of 1.01, 0.48, and 1.25 on CNN/DM, XSum, and WikiHow, respectively. The improvements on

| Method | Info. | Conc. | Read. | Faith. |
|---|---|---|---|---|
| BART | 3.78 | 3.10 | 3.25 | 4.74 |
| BART-Rewriter | **3.90** | 3.45 | 3.12 | **4.85** |
| GEMINI | **3.93** | **4.27** | **3.86** | **4.82** |

Table 5: *Human evaluation* on informativeness, conciseness, readability, and faithfulness on CNN/DM. GEMINI improves conciseness and readability by a large margin while maintaining other qualities.

ROUGE-L of CNN/DM and ROUGE-2 of Wiki-How are especially significant, reaching 1.44 and 1.56, respectively. The results suggest that the adaptive approach has an obvious advantage over the pure abstractive model.

Compared with the rewriter baseline BART-Rewriter, which relies on an external extractor to provide rewriting sentences, GEMINI improves the ROUGE-1/L scores by about 1.0 on CNN/DM, demonstrating the effectiveness of the adaptive approach compared with the pure rewriter. Compared with the HYDRASUM, which expresses the summary styles implicitly using a mixture of experts but obtains lower ROUGE scores than BART baseline, GEMINI achieves improved ROUGE scores using an explicit style control. The results demonstrate the advantage of a model with adaptive styles over pure rewriter and implicit style control.

Table 4 lists additional recent work using larger pre-training, assembling, reranking, and reinforcement learning techniques. However, these models are not directly comparable with our model. For example, PEGASUS (large) has 568M parameters, outnumbering BART (large) 400M by 42%. BRIO takes 20 hours per epoch and a total of 15 epochs to train on CNN/DM using 4 NVIDIA RTX 3090 GPUs, while our model only takes 2 hours per epoch and a total of 11 epochs to train on the same device, using only 7% computing resources of BRIO. We do not take these models as our baselines because they are in different lines of work using techniques perpendicular to this study. More importantly, we focus on the sentence-level summary style and its control instead of only ROUGE improvement.

## 6.2 Human Evaluation

We conduct a human evaluation to quantitatively measure the quality of generated summaries. We compare GEMINI with BART and BART-Rewriter baselines, studying four qualities, including *informativeness*, *conciseness*, *readability*, and *faithfulness*. We follow recent HYDRASUM (Goyal et al., 2022) and SummaReranker (Ravaut et al.,

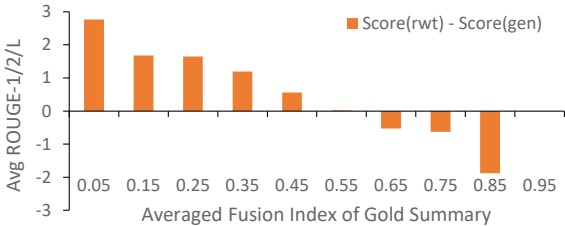

Figure 3: The rewriter (rwt) has higher average ROUGE scores than the generator (gen) on the buckets with low fusion index, evaluated on CNN/DM.

2022) to sample 50 documents from the test set of CNN/DM for the evaluation. We hire three graduate students with professional English proficiency (TOEFL scores above 100 out of 120) to annotate each candidate summary from 1 (worst) to 5 (best), and report the average scores across the three annotators.

As shown in Table 5, GEMINI achieves the best score overall, especially on conciseness and readability. Compared to BART-Rewriter, GEMINI gives close scores on informativeness and faithfulness, but higher scores on conciseness and readability. Compared to BART, GEMINI obtains better scores on the four metrics.

The explicit style control of GEMINI plays the role of a rough planner, which restricts the generation of content. Empirically, it generates summaries with a smaller number of sentences (3.3 sents/summary) than BART (3.9 sents/summary) and BART-Rewriter (3.7 sents/summary) on CNN/DM. As a result, GEMINI produces more concise summaries with lengths 20% shorter than BART and 10% shorter than BART-Rewriter but still obtains higher n-gram recalls as the higher ROUGE-1/2 in Table 4 shows, suggesting that the summaries generated by GEMINI tend to have denser information. We speculate the high readability of GEMINI is a result of its auto-regressive modeling of style prediction and sentence generation, through which the style transitions are optimized. We list two cases in Appendix A to illustrate the advantage of GEMIN on conciseness and readability.

## 6.3 Ablation Study

**Rewriter vs. Generator.** We further study our adaptive model by observing the contribution of the rewriter and the generator individually. We categorize the test samples in CNN/DM into 10 buckets according to the average fusion index, obtaining an averaged ROUGE-1/2/L score per bucket. We con-

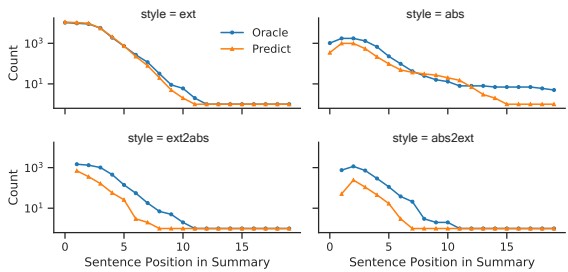

Figure 4: Distribution of *styles* and *style transitions* per sentence position in summary, evaluated on CNN/DM. The upper two sub-figures are about style distribution, while the under two are about style transitions. The sentence position denotes the ordinal number of the sentence in the summary.

| Method | R-1 | R-2 | R-L |
|---|---|---|---|
| CNN/DM | | | |
| GEMINI | 45.27 | 21.77 | 42.34 |
| – with random styles | 42.94 | 18.77 | 39.93 |
| – with oracle styles | **46.09** | **22.09** | **43.16** |
| XSum | | | |
| GEMINI | 45.86 | 22.55 | 37.68 |
| – with random styles | 45.71 | 22.43 | 37.56 |
| – with oracle styles | **45.94** | **22.70** | **37.79** |
| WikiHow | | | |
| GEMINI | 42.43 | 19.36 | 41.12 |
| – with random styles | 41.26 | 17.93 | 39.93 |
| – with oracle styles | **42.70** | **19.81** | **41.49** |

Table 6: Contribution of style prediction in comparison with random styles and oracle styles.

trast the ROUGE scores of the rewriter (rwt) and the generator (gen) as Figure 3. We can see that the rewriter dominates the performance on the region of low fusion index, while the generator rules the region of high fusion index. The distribution of ROUGE scores for the rewriter and the generator illustrates the specialization of the two styles.

**Pre-finetuning.** Pre-finetuning changes the distribution of randomly-initialized parameters. Take GEMINI on CNN/DM as an example. The initial average norms of embeddings for sentence identifiers and group tags are both 0.06, which are set purposely to a small value to reduce the negative impact on the pre-trained network. If we fine-tune the model directly, the trained model has average norms of 0.44 and 0.17 for sentence identifiers and group tags, respectively. However, if we pre-finetune the model for 8 epochs, the average norms climb to 0.92 and 0.63, respectively. After a follow-up fine-tuning, the average norms converge to 0.66 and 0.50, respectively, which are much higher than the model fine-tuned directly.

If we remove pre-finetuning, the performance of GEMINI on CNN/DM decreases from 45.27, 21.77, and 42.34 to 44.76, 21.60, and 41.71, respectively, on ROUGE-1/2/L, suggesting the necessity of the pre-finetuning stage.

## 7 Analysis

We try to answer two key questions related to sentence-level summary style control as follows.

### 7.1 Are sentence-level summary styles predictable?

We use oracle styles to train our model, which allows the model to be more easily trained. However, during testing, the summary style of each sentence can be arbitrarily chosen. It remains an interesting research question whether the style in the *test set* is predictable to some extent.

We try to answer this question using a quantitative analysis of style distribution on CNN/DM. We obtain an F1 of 0.78 for the style prediction, where more detailed distributions are shown as the upper sub-figures in Figure 4. The distribution of the predicted styles matches that of the oracle styles, except on a narrow range among sentence positions above 15, where the predicted abs style has less possibility than the oracle abs style. For the distribution of style transitions, as the under sub-figures show, the transitions ext-to-abs and abs-to-ext show some difference between the prediction and the oracle, where the predicted style transitions are overall less frequent than the oracle style transitions but with the same trends. The figures demonstrate the consistency in the distribution of predicted and oracle styles, suggesting that the styles are predictable to a certain degree.

We further evaluate the contribution of predicted styles by comparing the performance with the GEMINI decoding using randomly selected styles and oracle styles. As shown in Table 6, when we replace the predicted styles with random styles, the performance decreases by an average of 2.58 ROUGE points on CNN/DM and by an average of 1.26 on WikiHow. The performance drop indicates that the model's predictions of styles provide useful information to generate high-quality summaries. The prediction ability comes from the understanding of the input document. This also suggests that there is a consistent tendency in choosing the summary style of a summary sentence given its existing context.

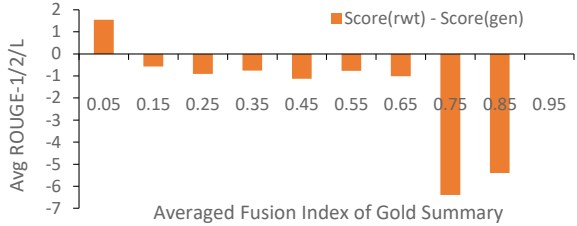

Figure 5: The rewriter (rwt) has higher average ROUGE scores than the generator (gen) on the buckets with a low fusion index, evaluated on WikiHow.

### 7.2 When will the adaptive model perform the best?

Intuitively, GEMINI works best on a dataset with a balanced style so that the rewriter and generator complement each other. More importantly, the oracle styles need to be accurate to obtain high-quality supervision signals.

First, the distribution of the dataset matters. Among the three datasets, WikiHow has the most balanced style which has 61.1% of the summary sentences preferring a rewriter. As a result, GEMINI obtains the most performance improvement on WikiHow by an average of 1.25 ROUGE points compared to the pure abstractive baseline, confirming our intuition about the relation between dataset distribution and model performance.

Second, complementary rewriter and generator are the essential prerequisites. As Figure 3 shows, the rewriter and generator on CNN/DM have relatively balanced capabilities, when the fusion index is below 0.55 the rewriter is preferred and above 0.55 the generator is preferred. In comparison, as shown in Figure 5, the rewriter and generator on WikiHow have skewed capabilities, where the rewriter is weak and only preferred when the fusion index is below 0.15. Consequently, GEMINI only generates ext-style for 19.3% of the summary sentences in contrast to the human assessment of 61.1%. The analysis suggests that GEMINI on WikiHow could be further enhanced by using a better rewriter, which could potentially be realized by using an improved sentence extractor.

Last, the quality of oracle styles determines the specialization of the two generators. As shown in Table 2, the Pearson correlation of the fusion index on WikiHow is only 0.56, which is much less than 0.76 on CNN/DM. It suggests a further improvement space on oracle styles by developing better automatic metrics.

## 8 Discussion

In this paper, we simulate human summary styles at the sentence level and provide a better understanding of how human summaries are produced. Based on this understanding, researchers could use GEMINI for different purposes. First, the automatic metric Fusion Index could be used to analyze the style distribution when developing a new dataset. Second, GEMINI could be used to control summary style at the sentence level. Since the ext-style summary sentences are naturally less likely to hallucinate than abs-style summary sentences, we could control the faithful risks by determining the proportion of abs-style summary sentences. Last, GEMINI produces an explicit style for each summary sentence, so we could even warn the faithful risks by marking those abs-style summary sentences in critical applications.

**Limitations.** GEMINI can fit the summarization style of a specific dataset so that the rewriter and generator specialize in different situations, which may enhance the quality of the rewriter and generator. However, we do not know if such an adaptive method actually improves the "abstraction" ability of a summary generation model. We do not have a reliable measure for the abstraction ability to evaluate the trained generator for now. We will consider it in our future research.

## 9 Conclusion

We investigated the sentence-level summary styles of human-written summaries, and evaluated the style distributions of the three benchmark datasets. We propose a fusion-index measure to detect sentence-level summary style automatically, using which we generate oracle styles for training our adaptive model. Experiments show that our GEMINI significantly outperforms pure abstractive and rewriter baselines on the benchmarks, demonstrating the effectiveness of fine-grained control in summary styles.

## Acknowledgments

We appreciate the constructive comments from the anonymous reviewers. We gratefully acknowledge funding from the Pioneer and "Leading Goose" R&D Program of Zhejiang (No.2022SDXHDX0003) and the National Natural Science Foundation of China (NSFC No.62161160339).

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

**Source:** Floyd Mayweather has a pop-up shop where he is selling T-shirts with a Philippines flag in the background . ' I like that , ' says Pacquiao with chuckle and then tells the leader of the Money Team : ' Welcome to the Manny team . ' We are accustomed to seeing boxers with grim expressions and gruff reactions - wearing what they call their fight face - in the last days before going into unarmed combat . Manny Pacquiao salutes the crowd as he makes his official arrival at the Mandalay Bay hotel . Pacquiao takes a selfie with his legendary trainer Freddie Roach as they greet the crowds . The Money Team t-shirts with the Philippines flag in the background on sale in Las Vegas . Pacquiao has a new camera which he points at everyone he comes across . Who do you think WILL win the fight between Floyd Mayweather and Manny Pacquiao ? ...

**Target:** Manny Pacquiao made his arrival at the Mandalay Bay hotel in Las Vegas . The Filipino takes on Floyd Mayweather at the MGM Grand on Saturday . The fight will be the richest of all time , grossing more than $ 300 million . Pacquiao is confident he can end Mayweather 's unbeaten record .

**BART:** Floyd Mayweather has a pop-up shop where he is selling T-shirts with the Philippines flag in the background . Manny Pacquiao is the happiest soul in Las Vegas , smiling , cracking jokes , bidding everyone welcome . His latest toy is a camera which he points at whoever he comes across , saying : ' I do n't know much about these things but I like taking portraits ' ' This is the right time for me to fight Floyd . I 'm more experienced now than five years ago . The speed and power are still there '

**BART-Rewriter:** Floyd Mayweather has a pop-up shop where he is selling T-shirts with a Philippines flag in the background . ' I like that , ' says Pacquiao with chuckle and then tells the leader of the Money Team : ' Welcome to the Manny team ' Pacquiao has a new camera which he points at everyone he comes across .

**GEMINI:** Manny Pacquiao was greeted by thousands of fans as fight week continues in Las Vegas . The Filipino boxer is confident he can beat Floyd Mayweather on Saturday night . Pacquio is the underdog going into the richest fight of all time against Mayweather .

---

**Source:** The Duchess of Cambridge is expected to go into labour any day now , but has a software designer predicted what the fourth in line to the throne will look like ? Programmer Charanjeet Kondal has used an application to create images of the new prince or princess . If he is right , the heir will have wispy blonde hair , dark brown eyes and a small nose when aged between two and four years old . Scroll down for video . Royal baby ? A software programmer has predicted the heir will have wispy blonde hair , dark brown eyes and a small nose when aged between two and four years old . The Duchess is expected to go into labour any day now , but has someone predicted how the baby will look . The pictures of the new prince or princess were created using Mr Kondal 's Morph Thing application . ...

**Target:** Software designer Charanjeet Kondal created images of new royal with app . If he 's right , heir will have blonde hair , dark brown eyes and a small nose . Betting public believe the royal baby will arrive into world on Wednesday .

**BART:** Programmer Charanjeet Kondal has used an application to create images of the new prince or princess . If he is right , the heir will have wispy blonde hair , dark brown eyes and a small nose . He ran the same software when it was announced Kate was pregnant for the first time with Prince George .

**BART-Rewriter:** Programmer Charanjeet Kondal has used an application to create images . He predicts heir will have wispy blonde hair , dark brown eyes and a small nose . The pictures were created using Mr Kondal 's Morph Thing application . He ran the same software when it was announced Kate was pregnant .

**GEMINI:** Programmer Charanjeet Kondal has used an application to create images of the new prince or princess . If he is right , the heir will have wispy blonde hair , dark brown eyes and a small nose when aged between two and four .

---

Table 7: Comparison of generated summaries from BART, BART-Rewriter, and GEMINI, evaluated on *CNN/DM*.

## A  Case Study

As Table 7 shows, the summaries generated by GEMINI are generally more readable and concise than the summaries generated by the BART and BART-Rewriter.