# OpenReview forum: "GEMINI: Controlling The Sentence-Level Summary Style in Abstractive Text Summarization"
_EMNLP/2023/Conference — EMNLP 2023 Main_

### Official Review · Reviewer_okNv · 2023-08-02

**Typos Grammar Style And Presentation Improvements:** 1. In line 059, I don't think it is f…
**Soundness:** 5

**Excitement:**

4: Strong: This paper deepens the understanding of some phenomenon or lowers the barriers to an existing research direction.

**Paper Topic And Main Contributions:**

This paper is about text summarization. They propose a model to switch between "extractive" and "abstractive" styles of summarization. For "extractive", the model rewrites one document sentence. For "abstractive", the model abstracts some information from the whole document.

To facilitate style switch, they propose a way to automatically detect the summary style for each summary sentence. Based on this, they build their GEMINI model on top of a pre-trained BART model and train the model with summary generation MLE loss and style prediction loss at the same time. Experiments show that GEMINI works better than BART and other strong baselines on 3 English benchmarks: CNN/DM, XSum, and WikiHow.

The contributions of this work are mainly made toward the NLP engineering experiment direction.

**Questions For The Authors:**

Question A: Could you elaborate on how group-tag embedding was used in the decoder? I find it a bit hard to understand by reading Figure 2 or the paper text. Do you add the group-tag embedding to input token embedding at each timestep for the decoder?

Question B: What loss did you use for pre-finetuning?

Question C: In section 6.1, have you checked the accuracy or F1 score for style prediction?

**Reasons To Accept:**

1. It is noval to switch between two styles for each summary sentence during generation. It combines two experts: rewriting one document sentence and abstracting information from a longer context.

2. The experiments are thoroughly done, and their claims are mostly well supported.

3. The paper is well-written and easy to follow.

**Reasons To Reject:**

1. The sample size (30 documents) of human evaluation is probably too small.

2. It is kind of contradictory between the claim that "WikiHow has the most balanced style, and thus GEMINI obtains the most improvement on WikiHow" and the fact that "GEMINI only generates ext-style for 19.3% of the sentences compared to the human labels of 61.1%".

3. No limitation section in the paper, which I believe is required.

4. One last personal and minor point: the way I view GEMINI is that it can fit the summarization style of a specific dataset very well but it may not actually improve the "abstraction" ability of a summary generation model. But what is the "abstraction" ability and how to evaluate it are questions I can't answer either. Hope this makes sense.

**Reproducibility:**

4: Could mostly reproduce the results, but there may be some variation because of sample variance or minor variations in their interpretation of the protocol or method.

**Reviewer Confidence:**

4: Quite sure. I tried to check the important points carefully. It's unlikely, though conceivable, that I missed something that should affect my ratings.

---

> ### Author Rebuttal · Authors · 2023-08-28
>
> Thank you for your constructive feedback.
>
> **Q1:** The sample size (30 documents) of human evaluation is probably too small. \
> **A1:** We will revise the paper to include more samples. We currently follow the previous BERT-EXT (Liu and Lapata, 2019) and BERT-Rewriter (Bao and Zhang, 2021) for the sample-size setting but we will increase it to 50 documents to match recent HYDRASUM (Goyal et al., 2022) and SummaReranker (Ravaut et al., 2022) in the revised version.
>
> **Q2:** It is kind of contradictory between the claim that "WikiHow has the most balanced style, and thus GEMINI obtains the most improvement on WikiHow" and the fact that "GEMINI only generates ext-style for 19.3% of the sentences compared to the human labels of 61.1%" \
> **A2:** We will revise the paper to clarify it. Compared to an imbalanced dataset such as CNN/DM, on which the generated abs-style is only 8.4%, the ratio of 19.3% of generated ext-style on WikiHow is relatively more “balanced”, which takes more advantage of the specialized rewriter and generator, and obtains higher performance improvement. However, the generated ext-style ratio of 19.3% is far less than the ratio of 61.1% of human-labeled ext-style, leaving a significant space for possible enhancement by improving oracle style labels and oracle rewriting sentences labels.
>
> **Q3:** The way I view GEMINI is that it can fit the summarization style of a specific dataset very well, but it may not actually improve the "abstraction" ability of a summary generation model. \
> **A3:** GEMINI can fit the summarization style of a specific dataset so that the rewriter and generator specialize in different situations, which may enhance the quality of the rewriter and generator. However, we do not have an reliable measure for the “abstraction” ability to evaluate our trained generator for now. We will consider it in our future research.
>
> **Q4:** Could you elaborate on how group-tag embedding was used in the decoder? I find it a bit hard to understand by reading Figure 2 or the paper text. Do you add the group-tag embedding to input token embedding at each timestep for the decoder? \
> **A4:** We will revise the paper to make it clear. The group-tag embedding is added to the input token embedding at each timestep for the decoder. When a decoder timestep predicts an identifier token, for example “<S3>”, the following timesteps will add the embedding of a group tag “3” until the end of the sentence “</S>”.
>
> **Q5:** What loss did you use for pre-finetuning? \
> **A5:** We will revise the paper to make it clear. The same MLE loss as fine-tuning is used during pre-finetuning. The only difference is that we freeze pre-trained parameters during pre-finetuning.
>
> **Q6:** In section 6.1, have you checked the accuracy or F1 score for style prediction? \
> **A6:** We will revise the paper to discuss it. On the CNN/DM test set, the F1 for style prediction is 0.78, where the precision is 0.79 and recall is 0.77.
>
> **Q7:** No limitation section in the paper, which I believe is required. \
> **A7:** We will discuss the limitation in the revised version. One limitation is that we do not explore the possibility of using large language model to mimic human summary styles, where if we could plan and dynamic switch such style using prompts other than model changes, the idea of controlling summary styles at sentence level should be more attractive.
>
> **Q8:** Typos and representation. \
> **A8:** We will fix them and carefully revise the paper to improve the clarity.

---

### Official Review · Reviewer_nZqP · 2023-08-02

**Soundness:** 3

**Excitement:**

3: Ambivalent: It has merits (e.g., it reports state-of-the-art results, the idea is nice), but there are key weaknesses (e.g., it describes incremental work), and it can significantly benefit from another round of revision. However, I won't object to accepting it if my co-reviewers champion it.

**Paper Topic And Main Contributions:**

The paper introduces a system named GEMINI for the task of textual summarization. The system consists of two complementary submodule: one more focused on the selection and rewriting of a single document sentence, and one focused on more abstrative generation of a novel sentence. GEMINI is tested on three standard summarization benchmarks (CNNDM, XSum, WikiHow), with results showing it leads to minor improvements over a simpler BART baseline, both in terms of ROUGE and a small-scale human evaluation.

**Questions For The Authors:**

See the questions listed above.

**Reasons To Accept:**

- The experimental setup is thorough, with experiments on three datasets, and an ablation study that confirms that there's a benefit to including both components in the system.
- The GEMINI model adds a layer of tracability/explainability to summarization, at least for the rephrasing component which can be more directly traced back in the document.
- The paper is generally well written, and comes together well in the Analysis Section which confirms that the gains of the model depend on the abstractiveness of the task.

**Reasons To Reject:**

- The human evaluation lacks depth and could be improved. It is not clear who was reruited to conduct the annotation, whether the scoring was anonymized, what the rater agreement level is, etc. The main result seem to be that GEMINI leads to better conciseness score, which is puzzling. The paper suggests that GEMINI summaries are shorter (which does not required human evaluation), but why is this the case?
- Since the human evaluation is minimal, the automatic results have more importance, yet recent work has shown the limited value of ROUGE scores, particularly in the day of RLHF, with work showing ROUGE and human preference are quite uncorrelated (Goyal, 2022). Is a gain of 1-2 ROUGE points valuable?
- The motivation overall for the work could be strengthened. Why the need for GEMINI? Is it to provide more user control, or better transparency? The minor ROUGE improvements (particularly when other models such as BRIO score much higher) seem limited. Who is the intended user of the GEMINI model?

**Reproducibility:**

4: Could mostly reproduce the results, but there may be some variation because of sample variance or minor variations in their interpretation of the protocol or method.

**Reviewer Confidence:**

4: Quite sure. I tried to check the important points carefully. It's unlikely, though conceivable, that I missed something that should affect my ratings.

---

> ### Author Rebuttal · Authors · 2023-08-28
>
> Thank you for your constructive feedback.
>
> **Q1:** The motivation overall for the work could be strengthened. Is it to provide more user control or better transparency? Who is the intended user of the GEMINI model? \
> **A1:** We will revise the paper to clarify it. We aim to mimic human summary styles, which we believe can both increase our ability to control the style and deepen our understanding of how human summaries are produced. Based on this understanding, researchers could use GEMINI for different purposes. First, the automatic metric Fusion Index could be used to analyze the style distribution when developing a new dataset. Second, GEMINI could be used to control summary style at the sentence level. Since the ext-style summary sentences are naturally less likely to hallucinate than abs-style summary sentences, we could control the faithful risks by determining the proportion of abs-style summary sentences. Last, GEMINI produces explicit style for each summary sentence, so we could even warn the faithful risks by marking those abs-style summary sentences in critical applications.
>
> **Q2:** The human evaluation lacks depth and could be improved. Who was recruited to conduct the annotation, whether the scoring was anonymized, what the rater agreement level is, etc. \
> **A2:** We will revise the part to clarify and include more analysis. We hire three graduate students with professional English proficiency (TOEFL scores above 100 out of 120). We ask them to annotate each candidate summary from 1 (worst) to 5 (best), where the scoring is anonymized. We report the score by averaging across the three annotators, where the rater agreement measured in ICC(3,1) is about 0.63, indicating a substantial agreement.
>
> **Q3:** Work shows that ROUGE and human preference are quite uncorrelated (Goyal, 2022). Is a gain of 1-2 ROUGE points valuable? \
> **A3:** We will add discussion in the revised version. Previous studies report that a summary with a high ROUGE does not necessarily have a high human preference, which indicates that the ROUGE score is a bad optimization target for obtaining good summaries. However, the ROUGE score as a quality metric still plays an irreplaceable role in text summarization tasks. When we compare the average ROUGE scores, especially when the test set is big, the ROUGE scores are still valuable indicators of the relative quality between models, where 1 ROUGE point usually indicates a significant quality improvement.
>
> **Q4:** GEMINI leads to a better conciseness score, which is puzzling, why is the case? \
> **A4:** We will revise the paper to discuss it. The explicit style control of GEMINI plays the role of a rough planner, which restricts the generation of content. Empirically, it generates summaries with a smaller number of sentences (3.3 sents/summary) than BART (3.9 sents/summary) and BART-Rewriter (3.7 sents/summary) on CNN/DM. As a result, GEMINI produces summaries with lengths 20% shorter than BART and 10% shorter than BART-Rewriter but still obtains higher n-gram recalls as the higher ROUGE-1/2 shows, which suggests that the summaries generated by GEMINI tend to have denser information.

---

### Official Review · Reviewer_EuEc · 2023-08-10

**Soundness:** 3

**Excitement:**

4: Strong: This paper deepens the understanding of some phenomenon or lowers the barriers to an existing research direction.

**Paper Topic And Main Contributions:**

This paper introduces a summarization technique that allows control over the sentence-level summary style. The authors achieve this by integrating both a rewriter and a generator. Notably, unlike prior research, they explicitly control the summary style by applying weak-supervised labels (either extractive or abstractive) to each sentence in the summary.

**Questions For The Authors:**

Question A: For a human evaluation, did you use the AMT? Are the three evaluators native speakers?

Question B: In Figure 4, What does the value of sentence position in summary (i.e., the x-axis value) mean?

**Reasons To Accept:**

The authors clearly describe the proposed model.

The ROUGE scores, obtained from three datasets, surpass those of the baselines.

A human evaluation was conducted to assess the quality of the generated summaries.

The ablation study effectively demonstrates the advantages of using the summary style controller.

**Reasons To Reject:**

While the results from both the automatic and human evaluations surpass the baselines, I am particularly concerned about the limited number of baseline comparisons. To my knowledge, there are additional baselines that employ BART-based models (e.g., GOLD[1] ...). It would be imperative for the authors to include comparisons with these works.

The authors have not clarified the distinction between the proposed model and BART-Rewriter. This omission may lead to confusion for readers regarding the unique contributions of this paper.

The authors have sampled only 30 documents from the CNN/DM test set, which, in my opinion, seems insufficient.
Also, the human evaluator's details are needed.

Unless I've overlooked something, the author should present more compelling evidence that aligning with human summarization styles results in the model producing more informative summaries. Such an alignment would ensure the summaries are genuinely beneficial for human readers. Conducting a human evaluation of the summary style on the generated summaries would be beneficial.



[1] PANG, Richard Yuanzhe; HE, He. Text Generation by Learning from Demonstrations. In: International Conference on Learning Representations. 2020.

**Reproducibility:**

4: Could mostly reproduce the results, but there may be some variation because of sample variance or minor variations in their interpretation of the protocol or method.

**Reviewer Confidence:**

4: Quite sure. I tried to check the important points carefully. It's unlikely, though conceivable, that I missed something that should affect my ratings.

**Typos Grammar Style And Presentation Improvements:**

It would be valuable to provide an explanation of baselines.

For me, Figure 2 is challenging to comprehend. It might be helpful if the author could provide a more detailed explanation or a clearer representation.

In my opinion, presenting the 'Related Work' section immediately after the 'Introduction' would enhance the paper's structure. This arrangement would help readers more easily discern the differences between implicit and explicit style control.

In Table 6, the results for random styles show a significant drop on the CNN/DM dataset— which predominantly features an extractive style—compared to other datasets. I believe this warrants further discussion.

---

> ### Author Rebuttal · Authors · 2023-08-28
>
> Thank you for your constructive feedback.
>
> **Q1:** There are additional baselines such as GOLD that employ BART-based models. It would be imperative for the authors to include comparisons with these works.\
> **A1:** We will add the comparison with GOLD. GEMINI achieves similar performance as GOLD on CNN/DM and XSum but with different and more likely perpendicular techniques. The RL training used by GOLD could potentially be applied to GEMINI to search for better ext/abs oracle labels, which should be able to further enhance the model performance.
>
> **Q2:** The authors have not clarified the distinction between the proposed model and BART-Rewriter, which may lead to confusion.\
> **A2:** We will revise the paper to clarify it. GEMINI has an explicit style control to dynamically switch between a rewriter and a generator. When it works as a rewriter, it selects a sentence internally for rewriting. In comparison, BART-Rewriter relies on an external extractor to provide the rewriting sentences, but the rewriting process is very similar to GEMINI.
>
> **Q3:** The authors have sampled only 30 documents, which seems insufficient. Also, the human evaluator’s details are needed. Are the three evaluators native speakers?\
> **A3:** We will revise the paper to include more samples and clarify the evaluation details. We follow the previous BERT-EXT (Liu and Lapata, 2019) and BERT-Rewriter (Bao and Zhang, 2021) for the sample-size setting but we will increase it to 50 documents to match recent HYDRASUM (Goyal et al., 2022) and SummaReranker (Ravaut et al., 2022) in the revised version. We hire three graduate students with professional English proficiency (TOEFL scores above 100 out of 120). We ask them to annotate each candidate summary from 1 (worst) to 5 (best) and report the average scores across the three annotators, where the rater agreement measured in ICC(3,1) is about 0.63, indicating a substantial agreement.
>
> **Q4:** The author should present more compelling evidence that aligning with human summarization styles results in the model producing more informative summaries.\
> **A4:** We will revise the paper to discuss it. Take CNN/DM as an example. GEMINI produces summaries with lengths 20% shorter than BART and 10% shorter than BART-Rewriter but still obtains higher n-gram recalls as the higher ROUGE-1/2 shows, which suggests that the summaries generated by GEMINI tend to have denser information.
>
> **Q5:** In Figure 4, what does the value of sentence position in summary (the x-axis value) mean?\
> **A5:** We will revise the paper to clarify it. The sentence position denotes the ordinal number of the sentence in the summary. For example, 1 – the first sentence, 2 – the second sentence, and so on.
>
> **Q6:** Typos and presentation.\
> **A6:** We will carefully revise the paper to improve the presentation, including explaining the baselines and Figure 2 more clearly, and rearranging the relate work and discussion for easier understanding.

---

### Meta-Review · Area_Chair_XJnv · 2023-09-19

**Recommendation:** 4

**Metareview:**

The paper explores methods for controlling sentence-level styles in abstractive summarization. The authors provide a clear description of their model, perform human evaluations to verify the summary quality, and support their claims with comprehensive experiments. However, reviewers have also raised concerns, including the lack of baseline comparisons with other methods, the insufficient sample size of 30 documents from the CNN/DM test set, the absence of details regarding human evaluation, the questionable reliability of ROUGE scores, among others.

---

### Decision · Program_Chairs · 2023-10-07

**Decision:**

Accept-Main

**Comment:**

The paper explores methods for controlling sentence-level styles in abstractive summarization. The authors provide a clear description of their model, perform human evaluations to verify the summary quality, and support their claims with comprehensive experiments. However, reviewers have also raised concerns, including the lack of baseline comparisons with other methods, the insufficient sample size of 30 documents from the CNN/DM test set, the absence of details regarding human evaluation, the questionable reliability of ROUGE scores, among others.